# The Impact of Socioeconomic Status on Adolescent Moral Reasoning: Exploring a Dual-Pathway Cognitive Model

**DOI:** 10.3390/bs15101347

**Published:** 2025-10-01

**Authors:** Xiaoming Li, Tiwang Cao, Ronghua Hu, Keer Huang, Cheng Guo

**Affiliations:** 1Faculty of Psychology, Southwest University, Chongqing 400715, China; lixiaoming2008@gues.edu.cn (X.L.); huangkeer@email.swu.edu.cn (K.H.); 2School of Foreign Languages, Guizhou University of Engineering Science, Bijie 551700, China; huronghua@gues.edu.cn; 3Student Affairs Office, Guizhou University of Engineering Science, Bijie 551700, China; caotiwang@gues.edu.cn

**Keywords:** socioeconomic status, moral reasoning, adolescent, social identity, cognitive flexibility

## Abstract

This study examines how objective (OSES) and subjective (SSES) socioeconomic status influence adolescent moral reasoning through distinct psychological mechanisms. Analyzing 4122 Chinese adolescents (Mage = 14.38), we found SSES enhanced moral internalization via strengthened social identity, while OSES reduced moral stereotyping through cognitive flexibility. Contrary to expectations, parental emotional warmth failed to buffer against SSES-related declines in internalization, with higher SSES predicting reduced internalization across parenting contexts. Results reveal socioeconomic status operates through dual pathways—social identity processes for SSES and cognitive flexibility for OSES—while challenging assumptions about parenting’s protective role. The findings suggest tailored interventions: identity-building programs for SSES-related moral development and cognitive training for OSES-linked reasoning biases, advancing theoretical understanding of moral development in diverse socioeconomic contexts.

## 1. Introduction

Adolescence constitutes a critical developmental stage characterized by rapid transformations in values, beliefs, and moral reasoning capacities. During this sensitive period, moral cognition matures and establishes enduring patterns that influence decision-making, psychosocial adjustment, and life-course development. Socioeconomic status (SES) has long been recognized as a salient contextual factor shaping adolescents’ moral development ([9]; [12]; [35]). Higher SES typically provides access to enriched educational opportunities, greater social capital, and supportive family contexts ([29]), which facilitate moral growth. Conversely, disadvantaged youth often face chronic stress ([22]), resource deprivation, and limited social support, constraining moral reasoning capacities ([43]). Thus, investigating the SES–moral reasoning nexus holds both theoretical and practical significance: it clarifies mechanisms underlying developmental disparities and informs equity-oriented educational and policy interventions.

### 1.1. Socioeconomic Status and Moral Reasoning

SES is a multidimensional construct typically measured through objective indicators—household income, parental education, and occupational prestige ([36])—as well as subjective SES (SSES), reflecting individuals’ perceived social standing ([27]; [38]). Distinguishing between OSES and SSES is crucial, as SSES often shows stronger associations with psychological outcomes such as well-being, self-esteem, and social cognition ([30]). Moreover, discrepancies between OSES and SSES can have significant implications: lower SSES relative to OSES predicts heightened stress, whereas higher SSES may promote resilience despite economic disadvantage ([34]).

Theoretical frameworks provide insight into SES effects on moral reasoning. Social Cognitive Theory emphasizes the role of observational learning ([4]): adolescents in high-SES environments benefit from exposure to prosocial role models, enriched discourse, and abstract ethical discussions ([47]). Conversely, low-SES contexts may necessitate pragmatic decision-making focused on immediate needs ([24]). [37]’s ([37]) stage theory further posits that socioeconomic conditions shape progression through moral stages: high-SES adolescents are more likely to achieve postconventional reasoning, while low-SES youth may remain at conventional levels emphasizing conformity, due to environmental stress and limited scaffolding ([8]). Empirical research supports these claims, demonstrating that higher OSES predicts more advanced reasoning in moral dilemmas ([9]) and that SSES independently shapes fairness and ethical evaluations ([26]). Parental education and social support further mediate these links ([47]).

### 1.2. Cognitive Flexibility as a Mediator

Cognitive flexibility—the ability to adapt thinking strategies to changing contexts—represents a key pathway linking OSES to moral reasoning. Higher-SES adolescents typically demonstrate stronger flexibility due to access to enriched learning opportunities, reduced chronic stress, and cognitively stimulating environments ([7]). This capacity enhances moral reasoning by enabling adolescents to (1) integrate multiple ethical perspectives, (2) generate alternative solutions, and (3) critically evaluate established norms ([15]; [10]; [17]). Flexible cognition allows for nuanced navigation of complex moral dilemmas, balancing individual rights with collective welfare. Importantly, high OSES provides material and educational resources—such as critical thinking curricula and extracurricular problem-solving—that foster this skill, thereby indirectly advancing moral reasoning.

### 1.3. Social Identity as a Mediator

Social identity, defined as one’s sense of belonging and psychological attachment to social groups ([57]), also serves as a mediator, particularly for SSES. Adolescents with higher SSES typically benefit from positive group affiliations, stronger networks, and inclusive identities, which promote moral norm internalization, perspective-taking, and concern for collective welfare ([54]; [58]; [52]). Conversely, lower SSES is associated with exclusion and fragmented identities, which may prioritize in-group loyalty at the expense of universalistic moral reasoning ([21]). Given that SSES directly shapes perceptions of belonging, its influence on moral reasoning through social identity may be even stronger than that of OSES ([28]).

### 1.4. Parenting Styles as a Moderator

Parenting behaviors—encompassing rejection, emotional warmth, and overprotection ([2])—represent proximal influences that condition SES effects on moral reasoning. Emotional warmth and authoritative parenting (high support, high expectations) consistently predict stronger moral reasoning ([14]; [49]), while rejecting or overprotective styles hinder it. Crucially, parenting practices can either buffer or exacerbate SES disparities: active involvement in low-SES households may compensate for limited resources ([60]), while universal emphases on empathy foster moral growth regardless of SES ([61]). Thus, parenting moderates the SES–moral reasoning relationship, with authoritative parenting serving as a protective factor even under disadvantaged conditions.

### 1.5. Study Aims and Hypotheses

The present study aims to clarify how SES—both objective and subjective—relates to adolescents’ moral reasoning through mediating and moderating mechanisms. Based on the reviewed literature, we propose the following:

**H1:** 
*Higher OSES will predict more advanced moral reasoning.*


**H2:** 
*Higher SSES will predict more advanced moral reasoning.*


**H3:** 
*Cognitive flexibility mediates the relationship between OSES and moral reasoning.*


**H4:** 
*Social identity mediates the relationship between SSES and moral reasoning.*


**H5:** 
*Parenting styles moderate the SES–moral reasoning relationship, with authoritative parenting buffering against SES-related disadvantages.*


## 2. Methods

### 2.1. Participant and Procedure

This study examined moral reasoning development among Chinese adolescents using a cross-sectional design. Participants were recruited from a stratified sample of 7th to 11th grade students across multiple secondary schools in Guizhou Province, China. The final sample consisted of 4122 adolescents (44.1% male, 55.9% female) with a mean age of 14.38 years (SD = 1.34). The grade distribution was as follows: 38.1% 7th graders, 43.3% 8th graders, 1.2% 9th graders, 7.2% 10th graders, and 7.5% 11th graders. Regarding family structure, 94.4% of participants reported having siblings while 5.6% were only children.

The research protocol followed ethical guidelines for educational research. Prior to data collection, active informed consent was obtained from both participants and their parents/guardians. Participation was voluntary with no compensation provided. Data collection occurred during regular school hours in designated computer classrooms, where participants completed a standardized battery of questionnaires assessing moral reasoning and related psychological constructs. The paper-based questionnaires were administered digitally using a secure online platform to ensure data integrity. Following completion, all responses were systematically organized and quality-checked by a trained research team.

This sampling approach yielded a demographically diverse adolescent population, with particular strengths in representing non-only children (reflecting China’s previous multi-child family norms) and capturing key developmental stages during the middle school years. The large sample size provides adequate statistical power for examining complex relationships between socioeconomic factors, psychological variables, and moral reasoning outcomes.

### 2.2. Measurements

The study employed well-established measurement tools with demonstrated reliability and validity across multiple studies. All instruments were carefully selected to ensure cultural appropriateness and measurement precision for the Chinese adolescent population.

Objective Socioeconomic Status (OSES): The SES Scale ([11]) was administered to assess family socioeconomic status through three key indicators: (1) parental education level (highest attainment), (2) parental occupational status, and (3) monthly household income. The 5-item scale utilized a 7-point Likert format with specific anchors: education levels ranged from 1 (illiterate/limited literacy) to 7 (master’s degree); occupation categories spanned from 1 (temporary worker/unemployed) to 7 (senior manager/technician); and income brackets progressed from 1 (≤600 RMB) to 7 (≥12,000 RMB). A composite SES score was calculated by summing responses across all items, with higher scores indicating higher socioeconomic status (α = 0.756).

Subjective Socioeconomic Status (SSES): The Chinese adaptation of the MacArthur Subjective Social Status Scale ([1]) measured adolescents’ self-perceived social standing using a 10-rung ladder metaphor. Participants indicated their family’s relative position in society (community subscale) and their personal status among school peers (school subscale). This visual analog scale has demonstrated strong validity in capturing subjective status perceptions across cultures.

Moral Reasoning: The Chinese version of the Prosocial Reasoning Objective Measure (CPROM; [40]) assessed moral reasoning through five vignettes depicting moral dilemmas: Blood Donation, Bullying, Accident, Swimming, and Flood. For each scenario, participants evaluated the importance of five reasoning dimensions (e.g., Hedonism, demand-oriented, stereotypes, recognition-oriented, internalization) using a 5-point scale. The measure yields a composite score reflecting moral reasoning sophistication (α = 0.899 across subscales).

Social Identity: The 8-item Social Identity Questionnaire ([31]) measured group identification using statements like “I strongly identify with certain groups” (7-point Likert scale; 1 = strongly disagree to 7 = strongly agree). The Chinese adaptation demonstrated good reliability (α = 0.914) in prior studies with adolescent samples.

Cognitive Flexibility: This study used the cognitive flexibility subscale from the Adolescent Executive Function Scale, which is suitable for adolescents ([32]). It consists of eight items and uses a three-point rating scale, with “1–3” representing “never,” “sometimes,” and “often,” respectively. The higher the total score, the less ideal the cognitive flexibility of the test subject. The measure captures both alternatives generation and control dimensions (combined α = 0.787).

Parenting Styles: The Simplified Parenting Style Scale (S-EMBU) was revised and simplified by [2] ([2]) on the basis of the standardized version of the parenting style scale (EMBU). Jiang revised the Chinese version of this questionnaire. The revised questionnaire was divided into two parts, the father’s and mother’s versions, with 21 questions in each part. The questions were consistent and had three relatively independent dimensions. The dimensions of rejection, emotional warmth, and overprotection were identified. The questionnaire was scored on a four-point scale, with 1 representing “never,” and 4 representing “always.” The internal consistency coefficient for this measure was 0.825.

All measures were administered in Mandarin following standardized translation/back-translation procedures. Scale reliabilities in the current sample remained consistent with established norms, supporting measurement validity. The comprehensive battery allowed for multi-dimensional assessment of key constructs while maintaining reasonable administration time (approximately 45 min).

### 2.3. Data Analysis

All analyses were conducted using SPSS 21.0 and the PROCESS macro. The analytic strategy proceeded in the following sequence:

Preliminary assumption testing: Data normality was assessed using skewness, kurtosis, and Kolmogorov–Smirnov tests. All variables approximated normal distribution, supporting the use of parametric tests. To check for common method bias, we conducted exploratory factor analysis (Harman’s single-factor test).

Bivariate associations: Pearson correlation analyses were performed to examine the zero-order relationships among SES (OSES and SSES), psychosocial mediators (cognitive flexibility, social identity), parenting styles, and moral reasoning. 

Regression analyses: Hierarchical regression models were used to test the predictive effects of SES on specific moral reasoning dimensions (internalization, demand orientation, stereotypes), while controlling for age. These analyses directly tested Hypotheses 1 and 2. Standardized β coefficients, *p*-values, and effect sizes (partial η^2^) are reported.

Mediation analyses: To examine indirect effects, we applied PROCESS Model 4 with 5000 bootstrap samples and bias-corrected 95% confidence intervals. Cognitive flexibility and social identity were tested as mediators of the OSES–moral reasoning and SSES–moral reasoning relationships, respectively.

Moderation analyses: To test whether parenting styles moderated SES–moral reasoning associations, we used PROCESS Model 1. Simple slope analyses were conducted for significant interactions.

This systematic approach allowed us to first establish baseline associations, then test predictive and indirect pathways, and finally evaluate interactive effects. Together, these analyses provided a comprehensive test of our hypotheses regarding the mechanisms linking SES and adolescent moral reasoning.

## 3. Results

### 3.1. Common Method Bias Test

The Harman single-factor test was conducted to assess common method bias through exploratory factor analysis. The analysis revealed that, under the non-rotated axis condition, 26 factors exhibited eigenvalues greater than 1. The first common factor accounted for 13.049% of the total variance, which is below the commonly accepted threshold of 40%. These findings therefore indicated the absence of common method biases in the study data.

### 3.2. Correlational Analysis Results

The correlation analysis revealed several significant relationships among key study variables (see Table 1). SSES demonstrated a moderate positive association with OSES (*r* = 0.284, *p* < 0.01), confirming their related yet distinct nature. Regarding moral reasoning outcomes, SSES showed a small but significant positive correlation with overall moral reasoning scores (*r* = 0.033, *p* < 0.05), with this relationship being particularly evident in the internalization (*r* = 0.041, *p* < 0.01).

Notably, SSES exhibited stronger associations with psychosocial variables than with moral reasoning measures. A significant positive correlation emerged between SSES and social identity (*r* = 0.132, *p* < 0.01), while a small but significant negative correlation was observed with cognitive flexibility scores (*r* = −0.10, *p* < 0.01). In terms of parenting dimensions, SSES correlated negatively with parental rejection (*r* = −0.067, *p* < 0.01) and positively with emotional warmth (*r* = 0.143, *p* < 0.01).

OSES demonstrated a different pattern of associations. While it showed no significant relationship with overall moral reasoning, it was negatively correlated with the demand-oriented (*r* = −0.044, *p* < 0.01) and stereotypes (*r* = −0.033, *p* < 0.05). Similarly to SSES, OSES was positively associated with social identity (*r* = 0.044, *p* < 0.01) and negatively with cognitive flexibility scores (*r* = −0.074, *p* < 0.01). The parenting style correlations mirrored those of SSES, with OSES showing negative associations with parental rejection (*r* = −0.101, *p* < 0.01) and positive associations with emotional warmth (*r* = 0.112, *p* < 0.01).

### 3.3. Regression Analysis Results

The prerequisite for regression analysis is to ensure the correlation between variables. Based on the results of the correlation analysis between socioeconomic status and moral reasoning level, we selected three dimensions of moral reasoning level for in-depth analysis. The results are shown in Table 2. The regression analysis was conducted to examine the predictive relationships between socioeconomic status (both subjective and objective) and specific dimensions of moral reasoning, while controlling for age. The results revealed distinct patterns of association for SSES and OSES.

For SSES, Model 1 demonstrated a significant positive prediction of moral internalization (*β* = 0.09, *p* < 0.05), supporting Hypothesis 1. This suggests that adolescents with higher self-perceived social status exhibit stronger internalization of moral principles, likely due to greater psychological security and social identification. In contrast, OSES (Model 4 and 6) showed significant negative predictions for both moral demand orientation (*β* = −0.224, *p* < 0.01) and moral stereotyping (*β* = −0.225, *p* < 0.05), thereby validating Hypothesis 2. These findings indicate that adolescents from higher socioeconomic backgrounds rely less on need-based or rigid moral reasoning, possibly due to greater exposure to diverse perspectives and abstract ethical discourse.

The differential effects of SSES and OSES highlight their unique roles in moral development: while SSES primarily influences the depth of moral internalization through self-concept and identity processes, OSES appears to shape the complexity of moral reasoning by providing structural resources and cognitive opportunities. These results underscore the importance of considering both subjective and objective socioeconomic measures when studying moral cognition in adolescents. Further implications of these findings are discussed in the following section.

### 3.4. Testing of Mediation Effects Results

To further elucidate the mechanisms underlying the relationship between socioeconomic status and moral reasoning, we conducted mediation analyses examining the roles of social identity and cognitive flexibility. The results revealed distinct mediating pathways for SSES and OSES.

For SSES (Figure 1A), social identity emerged as a significant mediator in the prediction of moral internalization (β = 0.077, Boot SE = 0.0038, 95% CI [0.0252, 0.0268]), supporting Hypothesis 3 and indicating that higher subjective status enhances moral internalization through strengthened group identification.

Conversely, for OSES (Figure 1B), cognitive flexibility significantly mediated the relationship with moral stereotyping (β = −0.018, Boot SE = 0.0087, 95% CI [−0.038, −0.004]), confirming Hypothesis 4 and suggesting that greater objective resources reduce rigid moral judgments by fostering cognitive adaptability.

Notably, social identity also mediated the OSES-moral demand orientation link (β = 0.042, Boot SE = 0.019, 95% CI [0.004, 0.079]), while cognitive flexibility showed no significant mediation effect in this pathway (95% CI [−0.004, 0.02]). These differential patterns demonstrate that (1) social identity serves as a broad mediator across socioeconomic dimensions, (2) cognitive flexibility operates more specifically in reducing stereotyped moral reasoning, and (3) distinct moral reasoning components engage separate psychological mechanisms. The complete mediation model, illustrated in Figure 1, substantiates our theoretical framework of dual socioeconomic pathways influencing adolescent moral development through both social identification and cognitive processing systems. These findings highlight the complex interplay between structural socioeconomic factors, psychological processes, and moral cognition during this critical developmental period.

### 3.5. Testing of Moderating Effects Results

Contrary to our initial hypothesis, the analysis revealed unexpected patterns in the moderating role of parenting styles. As presented in Model 3 (Table 1), while the interaction between SSES and parental emotional warmth showed statistical significance in predicting moral internalization (β = 0.012, *p* < 0.05), the direction of this relationship contradicted our theoretical expectations. Simple slope analysis (Figure 2) demonstrated that: Across all levels of parental emotional warmth, higher SSES was associated with decreased moral internalization. The negative relationship between SSES and moral internalization was slightly attenuated (but not reversed) under conditions of high emotional warmth. The protective effect of emotional warmth was insufficient to overcome the overall negative SSES–moral internalization association.

## 4. Discussion

The current study provides a comprehensive examination of how socioeconomic status (SES), through both objective (OSES) and subjective (SSES) dimensions, influences adolescent moral reasoning via distinct psychological mechanisms. Our findings reveal complex pathways through which socioeconomic factors shape moral development, while also yielding unexpected results that challenge conventional assumptions about parenting’s protective role.

### 4.1. The Dual Pathways of Socioeconomic Influence

Consistent with Social Cognitive Theory ([3]) and ecological models of development, our results demonstrate that OSES and SSES operate through different mechanisms. The mediation analyses confirmed the following.

SSES primarily enhances moral internalization through strengthened social identity (supporting H3), aligning with social identity theory’s emphasis on group belonging as a moral anchor ([57]). This suggests that adolescents’ self-perceived status may be more consequential than actual resources for internalizing moral values.

However, it is important to note that the MacArthur ladder measure of SSES may capture not only socioeconomic standing but also perceptions of peer acceptance and school-based social status. As such, the SSES–moral reasoning association might reflect adolescents’ integration into peer groups and their sense of social belonging as much as their perceived economic position. This nuance underscores the need to interpret SSES effects with caution and to consider its overlap with psychosocial constructs.

OSES reduces stereotyped moral reasoning via enhanced cognitive flexibility (supporting H4), validating [37]’s ([37]) premise that resource access enables advanced moral cognition. The negative mediation effect (−0.018) implies that economic advantages help overcome rigid moral judgments.

Notably, social identity also mediated OSES effects on demand-oriented reasoning—a novel finding suggesting that economic resources may indirectly foster moral development by enabling participation in identity-forming groups (e.g., extracurricular activities). This dual-mediation framework advances beyond traditional SES models by delineating when socioeconomic factors operate through social affiliative versus cognitive developmental pathways.

### 4.2. Counterintuitive Parenting Moderation Effects

Contrary to H5, emotional warmth failed to buffer against SSES-related declines in moral internalization. Instead, higher SSES predicted reduced internalization across all parenting levels (β = −0.12 to −0.08), with warmth offering only marginal attenuation. Several explanations merit consideration: (1) Status anxiety: High-SSES adolescents in competitive environments may prioritize achievement over moral internalization ([42]). (2) Peer influences: Affluent peer networks might emphasize instrumental over principled reasoning ([18]). (3) Measurement specificity: The PROM’s internalization scale may capture only certain aspects of moral reasoning.

In addition, the measure of parenting style used in this study (S-EMBU) emphasizes rejection, warmth, and overprotection, such as support for self-exploration, autonomy, and constructive risk-taking—dimensions that may be highly relevant for fostering adolescents’ moral reasoning. Future research should incorporate more comprehensive parenting frameworks to examine whether autonomy-supportive practices exert stronger protective effects across SES contexts. This unexpected result parallels recent findings on the “privilege paradox” ([48]), where socioeconomic advantages sometimes correlate with reduced ethical sensitivity.

### 4.3. Theoretical Implications and Practical Applications

Theoretical implications suggest needed refinements to dominant moral development theories. Beyond cognitive and contextual mechanisms, our findings align with broader perspectives emphasizing the interplay of cognition, personality, emotions, and personal beliefs in moral reasoning. Research demonstrates that emotional responses—such as guilt and empathy—play a crucial role in shaping ethical decision-making ([13]; [23]). These affective processes may operate independently of, or interact with, cognitive reasoning, suggesting that moral development models should integrate both rational deliberation and emotional sensitivity.

Neuroscientific evidence further underscores the importance of emotion in adolescence. The prefrontal cortex, responsible for higher-order reasoning, is still developing during adolescence, whereas the amygdala, involved in emotional processing, is relatively mature ([5]; [19]). As a result, adolescents may rely more on affective than logical processes in moral dilemmas ([33]). This developmental imbalance suggests that interventions focusing exclusively on cognitive skills may be insufficient; emotional regulation and socio-emotional learning are equally critical for fostering moral maturity.

Practically, our results recommend multi-dimensional interventions. For low-OSES adolescents, cognitive training can enhance flexibility and reduce rigid reasoning patterns, while for high-SSES adolescents, identity-based programs may counteract potential declines in moral internalization. Importantly, integrating socio-emotional learning into school curricula—through empathy-building exercises, reflective discussions on moral emotions, and guided peer interactions—can complement cognitive training. Parenting programs should combine emotional warmth with explicit moral discourse, ensuring that adolescents develop not only reasoning skills but also affective and identity-based resources for moral action ([51]; [25]; [56]).

Finally, an often-overlooked factor in moral development is birth order. Since the majority of our sample reported having siblings, it is plausible that being the oldest, middle, or youngest child may influence adolescents’ social responsibilities, identity roles, and even their engagement in moral reasoning ([46]; [59]; [45]; [53]).

### 4.4. Limitations and Future Directions

This study has several limitations. First, its cross-sectional design prevents causal inference ([39]). Second, focusing only on Guizhou adolescents restricts generalizability, as socioeconomic effects vary across contexts ([6]). Third, OSES relied on family-level indicators, neglecting neighborhood influences ([41]). Fourth, SSES may overlap with peer status ([27]). Fifth, parenting measures excluded autonomy support ([55]). Finally, birth order was not assessed, despite its potential relevance ([53]).

Future research should use longitudinal designs to establish developmental sequences ([20]), combine behavioral and self-report measures for ecological validity ([44]), and examine gene–environment interactions ([50]). Broader parenting assessments and multi-method approaches integrating behavioral, self-report, and neurocognitive tasks ([16]) would yield a more comprehensive picture and inform more effective interventions.

## 5. Conclusions

This study establishes that socioeconomic status influences adolescent moral reasoning through two distinct mechanisms. Subjective status (SSES) operates via social identity to enhance moral internalization, while objective status (OSES) functions through cognitive flexibility to reduce stereotyped reasoning. The surprising finding that parenting warmth did not mitigate SSES-related declines in internalization suggests socioeconomic advantages may sometimes hinder moral development. These results highlight the need for multidimensional approaches to moral education, including fostering inclusive social identities and cognitive adaptability. Future research should explore cultural and developmental moderators to better understand these complex relationships and inform targeted interventions addressing socioeconomic disparities in moral development.

## Figures and Tables

**Figure 1 behavsci-15-01347-f001:**
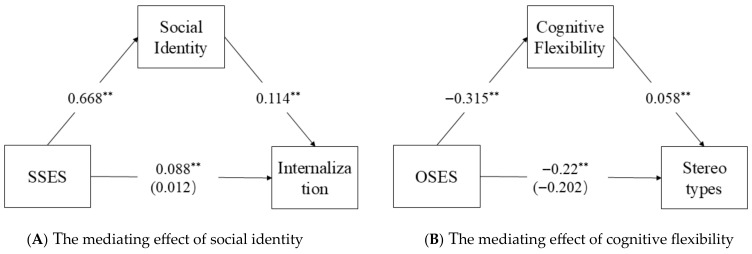
Mediating effect results. *Note*: ** *p* < 0.01.

**Figure 2 behavsci-15-01347-f002:**
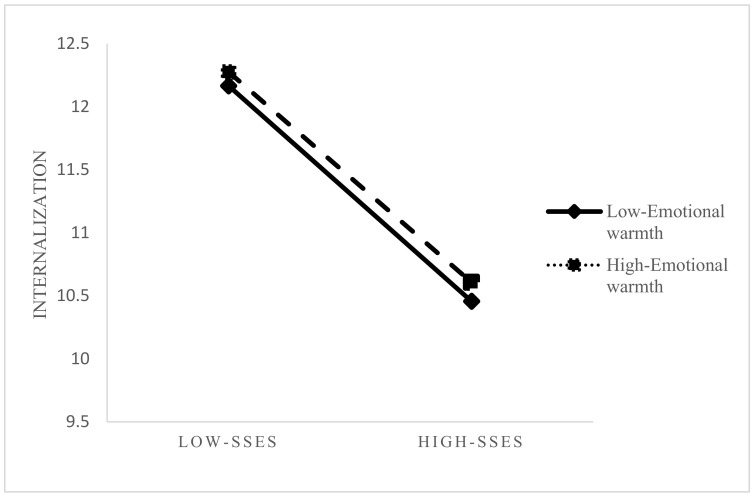
Moderation effect results.

**Table 1 behavsci-15-01347-t001:** Pearson Correlational analysis results (*N* = 4122).

	1	2	3	4	5	6	7	8	9	10	11	12	13	14	15
1. Sex	1														
2. Age	0.102 **	1													
3. SSES	−0.067 **	−0.146 **	1												
4. OSES	−0.058 **	−0.064 **	0.284 **	1											
5. Hedonism	−0.003	−0.028	0.004	−0.008	1										
6. Demand-oriented	0.029	−0.045 **	0.018	−0.044 **	0.383 **	1									
7. Stereotypes	−0.031 *	−0.116 **	0.025	−0.033 *	0.466 **	0.538 **	1								
8. Recognition-oriented	−0.002	−0.022	0.02	0.009	0.478 **	0.59 **	0.626 **	1							
9. Internalization	−0.002	−0.022	0.041 **	0.017	0.329 **	0.699 **	0.524 **	0.7 **	1						
10. Moral Reasoning	−0.003	−0.062 **	0.033 *	−0.014	0.566 **	0.833 **	0.796 **	0.822 **	0.886 **	1					
11. Social Identity	−0.012	−0.071 **	0.132 **	0.044 **	0.054 **	0.234 **	0.167 **	0.196 **	0.287 **	0.262 **	1				
12. Cognitive Flexibility	0.175 **	0.084 **	−0.1 **	−0.074 **	0.104 **	−0.018	0.049 **	0.031 *	−0.046 **	0.009	−0.183 **	1			
13. Rejection	−0.071 **	−0.04 *	−0.067 **	−0.101 **	0.131 **	−0.077 **	0.009	−0.01	−0.088	−0.038 *	−0.191 **	0.274 **	1		
14. Emotional warmth	−0.061	−0.113 **	0.143 **	0.112 **	0.072 **	0.262 **	0.162 **	0.188 **	0.284 **	0.267 **	0.396 **	−0.205 **	−0.034 **	1	
15. Overprotection	−0.036	−0.038 *	−0.025	−0.033	0.138 **	0.015	0.084 **	0.066 **	0.016	0.062 **	−0.091 **	0.267 **	0.607 **	−0.129 **	1

*Note*: * *p* < 0.05, ** *p* < 0.01.

**Table 2 behavsci-15-01347-t002:** Regression analysis results (*N* = 4122).

	Internalization			Demand-Oriented		Stereotypes	
	Model 1	Model 2	Model 3	Model 4	Model 5	Model 6	Model 7
Constant	17.604 ***	12.608 ***	11.897 ***	19.404 ***	15.332 ***	21.732 ***	21.001 ***
Age	−0.044	0.002	0.05	−0.112	−0.069	−0.346 ***	−0.359 ***
SSES	0.09 *	0.012	−0.544 *				
OSES				−0.224 **	−0.264 ***	−0.225 *	−0.202 *
Social Identity		0.114 ***	0.086 ***		0.086 ***		
Cognitive Flexibility							0.058 **
Emotional warmth			0.032				
Emotional warmth ∗ SSES			0.012 *				
*F*	3.804	107.319	93.259	6.664	67.789	24.282	18.837
*p*	0.022	0.000	0.000	0.001	0.000	0.000	0.000
*R*	0.044	0.276	0.333	0.062	0.236	0.118	0.127
*R* ^2^	0.002	0.076	0.111	0.004	0.056	0.014	0.016

*Note*: * *p* < 0.05, ** *p* < 0.01, *** *p* < 0.001.

## Data Availability

The data that support the findings of this study are available from the corresponding author upon reasonable request.

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
