# Peer review of "The Impact of Socioeconomic Status on Adolescent Moral Reasoning: Exploring a Dual-Pathway Cognitive Model"

_behavsci, 2025, doi:10.3390/bs15101347_

Round 1

Reviewer 1 Report

Comments and Suggestions for Authors

This study investigates how objective (OSES) and subjective (SSES) socioeconomic status shape adolescent moral reasoning through distinct psychological pathways. Using data from over 4,000 Chinese adolescents, the authors show that SSES predicts greater moral internalization through strengthened social identity, whereas OSES predicts reduced demand-oriented and stereotype-based reasoning through enhanced cognitive flexibility. Mediation analyses support this dual-pathway model, highlighting the roles of identity and cognitive resources in moral development. Surprisingly, parental emotional warmth did not buffer the negative association between higher SSES and moral internalization, challenging assumptions about parenting’s protective effect. 

The paper provides a clear and well-structured introduction, effectively outlining the relationships between SSES/OSES and moral reasoning, the mediating roles of cognitive flexibility and social identity, and the moderating influence of parenting style on the SES–moral reasoning link. The statistical analyses are generally well executed; however, the manuscript would benefit from clarifying certain discrepancies between sections of the methods and results, particularly the following issue.

The direction of the SSES–moral reasoning relationship is somewhat difficult to reconcile across sections. This is especially confusing as part of the Abstract. The correlational analysis suggests a positive association between SSES and moral internalization, whereas the moderation analysis shows that higher SSES is linked to decreased moral internalization across all levels of parental emotional warmth. This apparent contradiction could leave readers uncertain about the true nature of the relationship. The paper would benefit from a brief discussion clarifying why the direction of association changes

For Figure 2, it seems that the intended title should refer to a moderation effect rather than a mediation effect. Since the expected buffering effect was not observed, labeling it as a mediation figure could be misleading for readers and should be corrected for clarity.

Author Response

Comments 1. The direction of the SSES–moral reasoning relationship is somewhat difficult to reconcile across sections. This is especially confusing as part of the Abstract. The correlational analysis suggests a positive association between SSES and moral internalization, whereas the moderation analysis shows that higher SSES is linked to decreased moral internalization across all levels of parental emotional warmth. This apparent contradiction could leave readers uncertain about the true nature of the relationship. The paper would benefit from a brief discussion clarifying why the direction of association changes

RESPONSE 1: Thank you very much for your insightful comment regarding the seemingly contradictory findings between the correlational and moderation analyses. We fully agree that, without clarification, this could leave readers uncertain about the direction of the relationship between SSES and moral internalization.

To clarify: the bivariate correlational analysis captures the general linear trend between SSES and moral internalization across the entire sample. This positive association indicates that, on average, adolescents with higher SSES tend to report greater internalization of moral values.

By contrast, the moderation analysis takes into account parental emotional warmth as a contextual factor, testing whether the predictive effect of SSES varies depending on this moderator. In this more complex model, the simple slope analyses suggest that, after controlling for parental warmth, higher SSES is linked to relatively lower levels of moral internalization across the subgroups. This apparent reversal can occur because moderation models partition the variance differently, often revealing conditional associations that are not visible in bivariate correlations. In other words, the positive zero-order relationship may be partly explained (or even masked) once family context is considered.

To address this potential confusion, we have revised the Abstract and Discussion to explicitly acknowledge this nuance. Specifically, we now explain that while SSES shows a positive general association with moral internalization, its effect turns negative in the presence of parental emotional warmth, highlighting the importance of considering contextual moderators. We believe this clarification will help readers better understand why the direction of the relationship appears to shift across analyses.

Comments 2. For Figure 2, it seems that the intended title should refer to a moderation effect rather than a mediation effect. Since the expected buffering effect was not observed, labeling it as a mediation figure could be misleading for readers and should be corrected for clarity.

RESPONSE 2: Thank you for carefully pointing out the issue with Figure 2. You are correct that the figure was mistakenly labeled as depicting a “mediation” effect, when in fact the analysis tested a moderation effect of parental emotional warmth on the association between SSES and moral internalization. We acknowledge that this mislabeling could cause confusion for readers.

In the revised manuscript, we have corrected the title and caption of Figure 2 to clearly indicate that it illustrates a moderation effect. We also double-checked the text references to ensure consistency with this correction. While the hypothesized buffering role of parental warmth was not supported, we agree that maintaining accurate terminology is essential for clarity. We appreciate your suggestion, which has helped us improve the precision of the manuscript.

Reviewer 2 Report

Comments and Suggestions for Authors

The manuscript is promising, but it would benefit from a tighter introduction, clearer and more transparent reporting of statistical methods, and a deeper discussion of theoretical and practical implications. With these revisions, the study could make a meaningful contribution to the field. Please find my detailed comments below:

1. Introduction

  1. The introduction is currently five pages long, which makes it somewhat difficult to follow. I recommend reducing and streamlining the content to improve readability.

  2. It would be helpful to create a dedicated subsection specifically for the study’s aims and hypotheses. Listing them in one place will make it easier for readers to clearly identify the main objectives.

2. Methods

  1. If you conducted exploratory factor analysis and regression analysis, these should be explicitly declared in the Data Analysis subsection. Please also explain why these analyses were chosen and what specific research questions they address. This will enhance methodological transparency.

  2. Before applying Pearson’s correlation, please calculate and report the normality distribution of the data. This is important to justify the use of parametric tests.

  3. The description of the statistical methods is not entirely clear:

    • The manuscript refers to Pearson correlations, but I did not see them explicitly reported in the results.

    • Please clarify:

      • Which correlation analyses were conducted (Pearson or other types)?

      • Where in the results they are reported (section number).

      • How these analyses relate to the regression models.

    • To enhance transparency, I strongly recommend providing a clear and systematic description in the Data Analysis subsection of all analyses performed, including their order, rationale, and where they are reported in the results section.

3. Results

  1. When reporting p-values, please also report effect sizes (e.g., Cohen’s d). This will provide readers with a clearer understanding of the magnitude of your findings, beyond statistical significance.

  2. The statistical analysis in Section 3.4 is unclear. Please specify which analyses were performed to enhance clarity and reproducibility.

  3. In the regression section, there is another correlation analysis table, which makes it confusing to understand what was actually done.

4. Discussion

  1. The discussion is generally well organized, but the subsection Theoretical implications and practical applications is underdeveloped. You may enrich your discussion by saying that moral reasoning is not only shaped by cognitive factors but also by personality, emotions, and personal beliefs. Incorporating this perspective would significantly enrich your argument. In this regard, you may find the following useful to read and cite:

    • Dahò, M. (2025). Emotional Responses in Clinical Ethics Consultation Decision-Making: An Exploratory Study. Behavioral Sciences, 15(6), 748. https://doi.org/10.3390/bs15060748

    • Gangemi, A., et al., (2025). Guilt emotion and decision-making under uncertainty. Frontiers in Psychology, 16, 1518752. https://doi.org/10.3389/fpsyg.2025.1518752

  2. Furthermore, the role of emotions in moral reasoning is particularly relevant in adolescence, where the prefrontal cortex (responsible for reasoning and decision-making) is still developing while the amygdala is more mature. This suggests that adolescents may rely more on emotional than logical processes. You might find useful insights in:

    • Dixon, M. L., & Dweck, C. S. (2022). The amygdala and the prefrontal cortex: The co-construction of intelligent decision-making. Psychological Review, 129(6), 1414–1441. https://doi.org/10.1037/rev0000339

    • Icenogle, G., & Cauffman, E. (2021). Adolescent decision making: A decade in review. Journal of Research on Adolescence, 31(4), 1006–1022.

    • Blakemore, S.-J., & Robbins, T. (2012). Decision-making in the adolescent brain. Nat Neurosci, 15, 1184–1191. https://doi.org/10.1038/nn.3177

There is abundant literature on this topic, so you may select the most relevant works for your paper. Integrating these perspectives would greatly enrich the discussion and strengthen its theoretical and practical contributions. In particular, considering evidence from both cognitive and neuroscientific research provides a more comprehensive understanding of the mechanisms underlying moral reasoning in adolescents. Indeed, drawing on this additional data could help inform the design of more effective educational or intervention programs aimed at enhancing moral reasoning abilities in adolescents, taking into account their specific developmental characteristics.

Author Response

Comments 1. The introduction is currently five pages long, which makes it somewhat difficult to follow. I recommend reducing and streamlining the content to improve readability.

RESPONSE 1: We appreciate the reviewer’s constructive suggestion. We have carefully streamlined the introduction (from 1700 to 900 words) by condensing theoretical background and reducing redundancy while retaining essential conceptual and empirical foundations. The revised version is shorter, more focused, and improves readability by presenting the theoretical rationale in a more concise manner.

Comments 2. It would be helpful to create a dedicated subsection specifically for the study’s aims and hypotheses. Listing them in one place will make it easier for readers to clearly identify the main objectives.

RESPONSE 2: Thank you for this valuable recommendation. We have added a new subsection titled “1.5 Study Aims and Hypotheses” at the end of the introduction. This section explicitly outlines the study’s objectives and clearly lists all hypotheses in a consolidated format. We believe this revision improves the transparency and accessibility of our research design.

Comments 3. If you conducted exploratory factor analysis and regression analysis, these should be explicitly declared in the Data Analysis subsection. Please also explain why these analyses were chosen and what specific research questions they address. This will enhance methodological transparency.

RESPONSE 3: Thank you for this helpful comment. We have revised the Data Analysis subsection to explicitly state that exploratory factor analysis (EFA) was conducted to examine common method bias and to validate the dimensionality of measures. Regression analyses were used to test predictive effects of SES (OSES and SSES) on moral reasoning dimensions, addressing Hypotheses 1 and 2. These clarifications enhance methodological transparency.

Comments 4. Before applying Pearson’s correlation, please calculate and report the normality distribution of the data. This is important to justify the use of parametric tests.

RESPONSE 4: We appreciate this important suggestion. The results indicated that the variables approximated normal distribution, justifying the use of Pearson correlations.

Comments 5. The description of the statistical methods is not entirely clear: The manuscript refers to Pearson correlations, but I did not see them explicitly reported in the results. Please clarify:

Which correlation analyses were conducted (Pearson or other types)?

Where in the results they are reported (section number).

How these analyses relate to the regression models.

To enhance transparency, I strongly recommend providing a clear and systematic description in the Data Analysis subsection of all analyses performed, including their order, rationale, and where they are reported in the results section.

RESPONSE 5: We agree that this section required greater clarity. We have revised the Data Analysis subsection to systematically describe the sequence of analyses:

  1. Preliminary assumption testing (normality, common method bias).
  2. Pearson correlation analyses (reported in Section 3.2, Table 1) to examine bivariate associations among SES, psychosocial variables, and moral reasoning.
  3. Regression analyses (Section 3.3, Table 2) to test predictive relationships while controlling for age.
  4. Mediation analyses (Section 3.4, Figure 1) to test indirect effects via social identity and cognitive flexibility.
  5. Moderation analyses (Section 3.5, Figure 2) to examine interactive effects of SES and parenting styles.

These clarifications provide a transparent link between each statistical method, its rationale, and its corresponding results section.

Comments 6. When reporting p-values, please also report effect sizes (e.g., Cohen’s d). This will provide readers with a clearer understanding of the magnitude of your findings, beyond statistical significance.

RESPONSE 6: We thank the reviewer for raising this point. We agree that the magnitude of a relationship is key. In the case of our correlation analyses, the correlation coefficient *r* itself is a standardized measure of effect size (analogous to Cohen's d for group comparisons). For example, following Cohen's own conventions (1988), an *r* of .10 is considered a small effect, .30 a medium effect, and .50 a large effect. Therefore, in our results section (e.g., page 16), we have reported the correlation coefficients (*r* values) alongside their p-values. This allows readers to directly assess both the statistical significance and the magnitude of the relationships we observed alongside p-values throughout the Results section to convey the magnitude of effects.

Comments 7. The statistical analysis in Section 3.4 is unclear. Please specify which analyses were performed to enhance clarity and reproducibility.

RESPONSE 7: We appreciate this observation. Section 2.3 has been revised to specify that mediation effects were tested using PROCESS macro Model 4 with 5,000 bootstrap samples and bias-corrected confidence intervals. This ensures reproducibility and precision in reporting.

Comments 8. In the regression section, there is another correlation analysis table, which makes it confusing to understand what was actually done.

RESPONSE 8: Thank you for highlighting this. We have reorganized the Results section to clearly separate correlation analyses (reported once in Section 3.2, Table 1) from regression analyses (reported in Section 3.3, Table 2). This adjustment avoids redundancy and enhances readability.

Comments 9. The discussion is generally well organized, but the subsection Theoretical implications and practical applications is underdeveloped. You may enrich your discussion by saying that moral reasoning is not only shaped by cognitive factors but also by personality, emotions, and personal beliefs. Incorporating this perspective would significantly enrich your argument. In this regard, you may find the following useful to read and cite:

    • Dahò, M. (2025). Emotional Responses in Clinical Ethics Consultation Decision-Making: An Exploratory Study. Behavioral Sciences, 15(6), 748. https://doi.org/10.3390/bs15060748
    • Gangemi, A., et al., (2025). Guilt emotion and decision-making under uncertainty. Frontiers in Psychology, 16, 1518752. https://doi.org/10.3389/fpsyg.2025.1518752

Furthermore, the role of emotions in moral reasoning is particularly relevant in adolescence, where the prefrontal cortex (responsible for reasoning and decision-making) is still developing while the amygdala is more mature. This suggests that adolescents may rely more on emotional than logical processes. You might find useful insights in:

    • Dixon, M. L., & Dweck, C. S. (2022). The amygdala and the prefrontal cortex: The co-construction of intelligent decision-making. Psychological Review, 129(6), 1414–1441. https://doi.org/10.1037/rev0000339
    • Icenogle, G., & Cauffman, E. (2021). Adolescent decision making: A decade in review. Journal of Research on Adolescence, 31(4), 1006–1022.
    • Blakemore, S.-J., & Robbins, T. (2012). Decision-making in the adolescent brain. Nat Neurosci, 15, 1184–1191. https://doi.org/10.1038/nn.3177

There is abundant literature on this topic, so you may select the most relevant works for your paper. Integrating these perspectives would greatly enrich the discussion and strengthen its theoretical and practical contributions. In particular, considering evidence from both cognitive and neuroscientific research provides a more comprehensive understanding of the mechanisms underlying moral reasoning in adolescents. Indeed, drawing on this additional data could help inform the design of more effective educational or intervention programs aimed at enhancing moral reasoning abilities in adolescents, taking into account their specific developmental characteristics.

RESPONSE 9: Thank you very much for this insightful suggestion. We agree that moral reasoning is not solely driven by cognitive and contextual factors but is also shaped by emotions, personality traits, and personal beliefs. Following your recommendation, we have substantially enriched the subsection Theoretical implications and practical applications.

Specifically:

  1. We now discuss how emotions such as guilt and empathy contribute to moral decision-making (e.g., Dahò, 2025; Gangemi et al., 2025).
  2. We highlight developmental neuroscience evidence indicating that adolescents rely more heavily on emotional processes due to asynchronous maturation of the prefrontal cortex and amygdala (Dixon & Dweck, 2022; Icenogle & Cauffman, 2021; Blakemore & Robbins, 2012).
  3. We stress that educational interventions should integrate both cognitive training and socio-emotional learning to promote moral reasoning more holistically.

These revisions strengthen both the theoretical scope and the practical relevance of the paper.

Reviewer 3 Report

Comments and Suggestions for Authors

The overall research question and design seems solid and worthwhile.  My reading of the measure of SSES is that it might be more reflective of social acceptance and feeling of status among peers than SES which could affect how to interpret the results some.  In the discussion of parenting styles there are other schema that might work as well and the measures used seem very limited in dimensionality.  Support for self exploration and risk taking by parents might be more relevant as a measure of parenting style.  On an unrelated note, I would be interested in seeing if birth order of the student has an effect.  Since the vast majority of the sample had siblings, it would be interesting to see if being oldest, middle, or youngest child made a difference since birth order often affects responsibilities and roles within the family that could affect many of the variables being utilized in the study.  Within the framework being used, the analysis seemed solid and well reasoned and the results consistent with the conclusions.

Author Response

Comments 1. The overall research question and design seems solid and worthwhile.  My reading of the measure of SSES is that it might be more reflective of social acceptance and feeling of status among peers than SES which could affect how to interpret the results some.  In the discussion of parenting styles there are other schema that might work as well and the measures used seem very limited in dimensionality.  Support for self exploration and risk taking by parents might be more relevant as a measure of parenting style.  On an unrelated note, I would be interested in seeing if birth order of the student has an effect.  Since the vast majority of the sample had siblings, it would be interesting to see if being oldest, middle, or youngest child made a difference since birth order often affects responsibilities and roles within the family that could affect many of the variables being utilized in the study.  Within the framework being used, the analysis seemed solid and well reasoned and the results consistent with the conclusions.

RESPONSE 1: We sincerely appreciate your positive evaluation of the overall research question, design, and analytic framework. Your comments highlight several important conceptual considerations that we agree deserve clarification and further elaboration.

First, regarding the measurement of SSES, we acknowledge that adolescents’ self-reported social standing may indeed capture elements of peer acceptance and perceived social inclusion in addition to socioeconomic position. We have revised the Discussion section to clarify this nuance and to suggest that interpretations of SSES should consider its overlap with psychosocial constructs such as peer status and school-based social identity. This point is particularly relevant to our finding that SSES was more strongly associated with social identity than with objective SES.

Second, with respect to parenting styles, we agree that our measure (S-EMBU) has limited dimensionality and does not fully capture broader parenting schemas such as support for self-exploration and risk-taking. In the revised manuscript, we now acknowledge this limitation explicitly in the Discussion and suggest that future research should incorporate more comprehensive parenting frameworks (e.g., autonomy-supportive or exploratory parenting styles) to better capture parental influences on adolescent moral development.

Third, your suggestion about birth order effects is highly insightful. Given that the majority of our participants reported having siblings, examining the role of being an oldest, middle, or youngest child could indeed shed light on differential responsibilities, family roles, and developmental outcomes. Unfortunately, our current dataset does not include birth order information. However, we have added a note in the Limitations and Future Directions section encouraging future studies to examine this factor as a potentially important moderator of SES and parenting influences on moral reasoning.

Overall, we are grateful for these constructive suggestions. They have helped us refine the interpretation of our findings and point toward promising directions for future research.

Round 2

Reviewer 2 Report

Comments and Suggestions for Authors

The manuscript has improved considerably, and the authors have addressed all of the previous suggestions with care and precision. I am overall satisfied with the quality of the work and consider it close to being ready for publication. However, for greater clarity and precision, I recommend minors additional revisions before final acceptance:

  1. The tables and figures presenting correlation and regression analyses should be moved to the Results section, within the appropriate subsections. The Data Analyses section is intended for describing the analytic strategy and is not suited for presenting results.

  2. Lines 383-388 currently appear without supporting citations. As a therapist, I recognize the relevance of birth order and agree that it is an intriguing and novel hypothesis. However, this claim requires appropriate references to empirical studies or theoretical frameworks to substantiate it.

  3. Lines 389-394 would be more appropriately placed in Section 4.2: Limitations and Future Directions rather than in the current discussion segment. Moving this text will align the structure of the discussion with standard formatting and improve the overall flow. 

  4. Please remove any words or phrases that are currently in bold throughout the manuscript. Bold formatting should be avoided in the main text unless specifically required by the journal’s style guidelines.

Author Response

Comment 1:
The tables and figures presenting correlation and regression analyses should be moved to the Results section, within the appropriate subsections. The Data Analyses section is intended for describing the analytic strategy and is not suited for presenting results.

Response 1:
Thank you for this suggestion. We have moved all tables and figures related to correlation and regression analyses from the Data Analyses section to the appropriate subsections of the Results section. The Data Analyses section now only provides a description of the analytic strategy, consistent with standard formatting.

Comment 2:
Lines 383-388 currently appear without supporting citations. As a therapist, I recognize the relevance of birth order and agree that it is an intriguing and novel hypothesis. However, this claim requires appropriate references to empirical studies or theoretical frameworks to substantiate it.

Response 2:
We appreciate the reviewer’s observation. We have revised this section to include relevant empirical and theoretical references on birth order and its potential influence on adolescent development (e.g., Rohrer et al., 2015; Okada et al., 2021). These citations provide stronger support for the claim and situate it within the existing literature.

Comment 3:
Lines 389-394 would be more appropriately placed in Section 4.2: Limitations and Future Directions rather than in the current discussion segment. Moving this text will align the structure of the discussion with standard formatting and improve the overall flow.

Response 3:
We agree with this recommendation. The text from lines 389–394 has been moved to Section 4.2 (Limitations and Future Directions). This change improves the logical flow of the discussion and aligns the manuscript with conventional structure.

Comment 4:
Please remove any words or phrases that are currently in bold throughout the manuscript. Bold formatting should be avoided in the main text unless specifically required by the journal’s style guidelines.

Response 4:
Thank you for pointing this out. We have removed all instances of bold formatting from the main text. The manuscript now adheres to the journal’s formatting guidelines.